# The Effect of Gender and Age in Small Bicycle Sharing Systems: Case Study from Logroño, Spain

**Alexandra Cortez-Ordoñez** [1] and **Ana Belén Tulcanaza-Prieto** [2,*]

1 Departamento de Estadística e Investigación Operativa, Universidad Politécnica de Catalunya, 08034 Barcelona, Spain; alexandra.cortez@upc.edu
2 Escuela de Negocios, Universidad de Las Américas, UDLA, Quito 170124, Ecuador
* Correspondence: ana.tulcanaza@udla.edu.ec

**Abstract:** During recent years, bike sharing systems (BSS) have been adopted in many large cities around the world. Thanks to their environmental and health benefits, BSS' popularity as a green transportation mode is exponentially increasing and many small cities are also adopting them. However, few of these small cities have the resources to manage and analyze the massive amount of data produced by these systems in order to optimize them and promote their use among citizens. This manuscript analyzes BiciLog (Logroño, Spain) data and studies customers' usage patterns, disaggregated by gender and age. The *t*-test is the inferential statistic test employed to compare the equality of the means among different groups. Results show differences in how women and men are using the BiciLog system. Women use the system less but ride for longer than men. There are also differences between age groups. Most of the users are between 20 and 29 years old. However, customers between 60 and 69 years old are also extensively using BSS. In fact, they not only make more trips but also their rides are around three times longer than customers in other age groups. These results can be used by BiciLog operators to create and evaluate campaigns to motivate BSS use among target groups and improve the system based on customers' preferences. The main limitation of this investigation is the lack of data available to calculate additional information such as the real distance covered by customers when riding, or their preferred routes. For future research, a longer data period can be considered to compare usage patterns across different years. Additionally, customer surveys can help us to understand their motivations to use the system and corroborate the results found in this study.

**Keywords:** BSS usage patterns; bike usage analysis; gender; age; inclusive transport system; new modes of transport





## 1. Introduction

Bicycle sharing systems (BSS) were initiated by local community organizations and charitable projects to promote bicycles as a non-polluting form of transportation. These systems were implemented using the concept of mobility as a service. Specifically, BSS started as a public service in 1965 in Amsterdam with the program "white bike" [1]. However, BSS have become a popular transportation mode in recent years thanks to their multiple environmental and economic benefits. Moreover, BSS provide advantages in the social and health aspects, such as pollution reduction and an improvement in citizens' health, and a new and convenient transportation mode that is low-cost and environmentally friendly [2–4]. BSS are commonly used as a replacement for other transportation means for short- or medium-distance trips, and for the first-(or last-) mile connection to other transportation modes. Currently, there are more than 1700 BSS around the world [5] of different types (public, private, dock less) and of different sizes. In fact, BSS have been presented in larger cities such as Paris or Barcelona for more than one decade, and there are older systems, such as the one in Toronto. Smaller cities have also been adopting

BSS [6]. Their numerous benefits (e.g., less congestion, low transport exclusion, high quality of transportation connections, improvement of the quality of life of the inhabitants, and increase in the image and innovativeness of the town), have been especially noted in the small- and medium-sized towns of the Łódź region in Poland, as is mentioned in Kwiatkowski's study [7]. The author also mentioned that BSS develop tourism and recreation in small towns because BSS can be shown as the first mode of public transport for inhabitants and foreigners [7]. In Spain, these systems are also becoming popular, and small cities such as Logroño, Girona or Salamanca, provide their citizens with this alternative mode of transportation. BSS have been recognized as an important tool towards the construction of sustainable cities by the reduction of the carbon footprint; therefore, the systems have experienced an unprecedented expansion worldwide. BSS also help in the fight against mobility poverty given their low price compared to ownership; however, the requirement to pay with credit or debit card could be a major impediment for the population with less income. Combining card and cash payments could lower the barriers for those social groups.

The use of BSS is generally complemented by mobile applications or web services where customers can check the status of the systems and of particular stations. However, there is important information that is still missing, such as availability prediction and bike booking in advance, among others. On the other hand, local governments who provide the service also need to monitor it. Usually, a contractor, a third part company or a small team in the local government provide them with this information. Nevertheless, these data are usually provided as highly aggregated reports or as raw data. Additionally, both customers and managers of the system oftentimes complain that the data are not reliable enough or not updated frequently. The issue of logistics is also an important question in the BSS area. Data to identify spatial-temporal patterns and predict cycle demand are necessary for users and operators to make more informed decisions. Moreover, there are challenges for the design of modern algorithms, visualization systems, and machine learning techniques to provide possible solutions to the rebalancing problem (e.g., polluting vehicles, other vehicle-sharing systems, personalization of the service, topology, socio-economic factors, climatology) [8].

To effectively analyze public BSS there are several variables to consider, such as the size of the system (small, medium, or large), the quality of the data, or the type of user on which to place the focus (customers or managers of the system). As mentioned before, thanks to their environmental and health benefits, BSS' popularity as a green transportation mode is exponentially increasing and many small and medium cities are also adopting them. In fact, worldwide, most of them are medium to small. For instance, in Spain, with around forty systems, only five of them have more than seventy docking stations. However, few of these small cities possess the resources to manage and analyze the massive amount of data produced by these systems in order to optimize them and promote their use among citizens. With the purpose of concentrating on the most common system size, a collaboration with Logroño City Hall was established. Logroño is a city located in the north of Spain (see Figure 1) with a population of 150 k inhabitants. Their BSS, called BiciLog, has currently 25 docking stations installed around the city.

The data quality is also another aspect to consider. Larger cities provide free access to the data through Open Data websites, such as the one from Barcelona [9]. Smaller cities, such as Logroño, do not provide open data. However, as part of the study's collaboration, Logroño's City Hall provided access to the original data sources. Finally, considering the type of user, the focus will be on the managers and system operators (from now on called supervisors) of Logroño's BSS.

The main objective of this study is to promote a deep analysis of Logroño's citizens, who are users of the BSS (called customers), and provide BiciLog supervisors with a high-level understanding of customers' motivations to participate in these sustainable and environmentally friendly programs. The study focuses on BSS customers' usage patterns

according to their gender and age and tests if there are differences in how these groups are using the system.

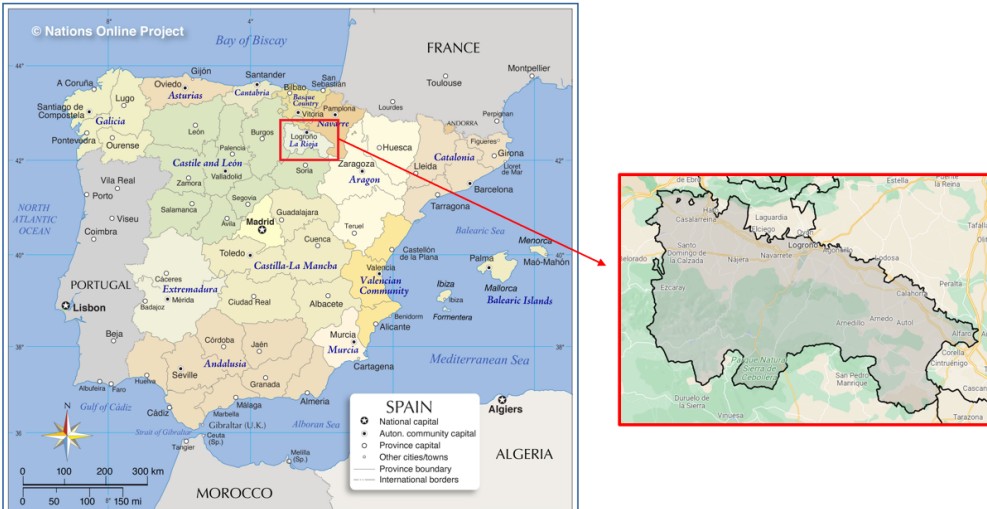

**Figure 1.** Spain Administrative map. Logroño is located in the north of the country, in La Rioja province. Source: Nations Online Project and Google maps.

The remainder of the article is organized as follows. Section 2 presents the literature review and describes the hypothesis development, Section 3 focuses on the data collection procedure and methodology design, Section 4 presents the empirical findings, and Section 5 presents the discussion. Finally, Section 6 outlines the conclusions.

## 2. Literature Review and Hypotheses Development

Most BSS enable users to access bikes at any hour on an as-needed basis from a network of stations [10], which are typically located in urban areas. Other BSS are docking-free and customers can pick up and drop off bikes at random places in the urban areas. These schemes might facilitate the use of cycling rather than other non-sustainable modes of transport. Moreover, the success rate of BSS is varied since many factors influence their prosperity, including demographics, local geography, and users' preferences. Specifically, [10] distinguished between the following types of BSS:

- Public BSS: Schemes where anyone is able to access a bicycle for a nominal fee using a credit or debit card.
- Closed-campus BSS: Schemes developed by universities and office campuses, which are only available to the particular campus or community that they serve.
- Person-to-person (P2P) BSS: Schemes available in urban areas for bike owners to rent out their bikes for others to use.

There are several pros of BSS. Bieliński et al. [11,12] studied the impact of the new electrically assisted bicycle (e-bike) sharing system in Northern Poland. Their findings suggested that e-bike sharing is a competitive system compared to carsharing, moped, and taxi services. However, it only substitutes and supplements public transportation. Woodcock et al. [4] performed a study of the London BSS and their effects on health. The authors concluded that London BSS have a positive health impact overall. However, these benefits are differentiated by gender and age. Therefore, these advantages are higher for men than for women and for older users than for younger users. Moreover, Fishman [13] mentioned that the potential benefits depend on which travel mode is replaced by BSS. The author indicated that traffic safety is a crucial variable because bike sharing may even be safer than riding a private bike. Generally, the design of bicycles used in shared systems is complete and integrated, while bike sharing users are more cautious than other

public/private users. Therefore, some countries evaluate the possibility of providing BSS users with price incentives to redistribute bikes.

The use of BSS can be studied in different ways. For instance, focusing on the influence of urban configuration [14–16], the traffic patterns [17–19] or the characteristics of trips [20–22]. These studies shared with the authors the objective of understanding how BSS are being used. Nevertheless, this study focuses on the analysis of customers' patterns according to their gender and age rather than identifying the demographic factors that could influence bike usage flows. Moreover, the effects of important events [18], the weather [23,24] or calendar events [24–26] have also been important factors for studying the usage of BSS in recent years. These studies identify the external factors that can influence BSS adoption, but no analysis of the customers' behavior patterns has been conducted, which is the main contribution of this study.

Identifying the usage flows is a key factor for optimizing rebalancing operations [27–31]. Given that in Logroño the rebalancing of stations' operations is conducted by a third-party company and the current study lacks access to their routes, this manuscript will only focus on the analysis of usage patterns using customers' information. Several visualization systems have been developed to analyze BSS from different angles, mainly focusing on the stations' statuses. For instance, the difference between weekdays and weekends [32], dynamics across stations [33] or tracking several systems across the world [5]. This study will not provide any visualization system. However, the results of this study can be easily incorporated into the visual tool developed by Cortez-Ordoñez et al. in [34]. Fusing BSS usage data with other transportation means [35] also provides a good understanding of the stations that can have outages during certain peak hours. Unfortunately, these data are not available in the case of Logroño, and it is out of the scope of this study.

Considering that the number of stations in Logroño is small and that the study focusses on the understanding of BiciLog usage based on customers' characteristics (gender and age), there is no need to cluster the data as other authors have done previously [33,36,37]. Similarly, the prediction of bike flows has been a hot topic in recent years and has been developed by several authors [17,38–40]. However, this study focuses on the usage patterns with the customers' information and does not predict the usage flows.

Prior studies analyzed BSS using different disaggregation techniques according to the research objectives. Beecham et al. [41] study the commuting behaviors of users by analyzing the journey data to gain insights into the people using those services and the geography of commuters' workplaces. Specifically, studying BSS usage by gender and group age generated different findings. Women use the BSS less than men in all the tested time slots of the study performed in Valencia [42], which also showed different network density and centrality between genders, concluding the presence of gender inequality patterns of use and space occupation. Moreover, a study performed in the largest public BSS in the U.S. (Bluebikes in Boston, City Bike in New York, and Divvy Bikes in Chicago) showed that the portion of trips made by women increased over 22% in all programs in the period of 2014–2018 [43]. However, in all schemes, the gender gap was higher for older bicycle share users. Pans et al. [44] indicated that women use the Valencia BSS less than men, and older people employ the BSS more frequently than younger people, while people with a lower socio-economic level have a higher rate usage than people with a high and medium socio-economic level. Moreover, a study performed in the Paris Region of Petite Couronne (France) established that women used BSS much less than men (about 30% of the total number of users) [45]. Women were concerned about: accessibility (availability of bikes, distance to the nearest station, type and quality of the cycle paths), safety and security, social constraints, weather, and topography. Finally, women were most affected by the COVID-19 pandemic as they are more likely to be dependent on public transportation, which was heavily restricted during 2020 and 2021. Particularly, a BSS study conducted in Lisbon [46] revealed that men have higher bike ownership rates and combine the personal use of a bike with BSS. Conversely, women more frequently integrated BSS and public transport.

Furthermore, while men were using BSS more regularly than women pre-pandemic, during the COVID-19 pandemic, women were using BSS as recurrently as men.

Therefore, the hypotheses of the study are the following:

**Hypothesis 1 (H1)**. *There are significant differences in BiciLog usage by gender*.

**Hypothesis 2 (H2)**. *There are significant differences in BiciLog usage by age groups*.

## 3. Research Model

### 3.1. Data Collection

The information from Logroño's BSS is not publicly available. Therefore, the authors of the study requested access to data through Logroño's City Hall. The data used in this study were provided while respecting the European Data Protection Laws in the Spanish Reglamento General de Protección de Datos (GDPR). Moreover, the BiciLog data of customers as well as real-time trips from October 2019 are available on a private website that is maintained by the Instituto Tecnológico de Castilla y León (ITCL).

Even though the data are available from October 2019, the selected period of analysis for this study is from January 2021 to December 2022. This way, it is possible to compare how usage behavior has changed in the last two years. Moreover, during 2020, there were several mobility restrictions as a consequence of the COVID-19 pandemic. Therefore, customers drastically changed their usage patterns. Since 2021, the restrictions have been lifted gradually, creating a new usage behavior in BiciLog customers, which will be analyzed and compared in this study. Table 1 presents the variables involved in the trips and customer data file. During the considered period, the customer data file had 3834 records while the trips file had more than 48k records.

**Table 1.** Trips and customer data description.

| Trips File Variables | Customer File Variables |
| --- | --- |
| Bike ID | User ID |
| Start station ID | Is Man (yes/no) |
| Start docking ID | Is Woman (yes/no) |
| End station ID | Date of birth |
| End docking ID | Account creation date |
| Start date and time | Is maintenance user (yes/no) |
| End date and time | City |
| User ID | Province |
| | Country |

Source: BiciLog private website.

The first step is to clean the trips data file. Logroño's BSS has 25 docking stations distributed across the city (see Figure 2). From those, three stations were added to the system in the last quarter of 2021 and two stations in the last quarter of 2022. These stations are not considered as part of the analysis because there is not enough historical information to perform a yearly comparison. Removing these stations represents around 4% of the total dataset. Records of stations that were temporally non-working and bike loans that did not have an ending time frame or an end station registered were also removed from the dataset (less than 0.5%). The Bike ID and start and end docking ID columns were also removed as they were not used in this analysis.

On the other hand, the data provided in the customer's file come from the information customers registered when they created their user account. As mentioned before, personal data are not considered in this study as this is not under the scope of GDPR. Garbled data such as maintenance customers (less than 0.5%) and others that were missing important information such as date of birth or gender (less than 4%) were removed from the dataset. Other features such as Account creation date, City, Province, and Country were also removed from the dataset as they were not used in the analysis.

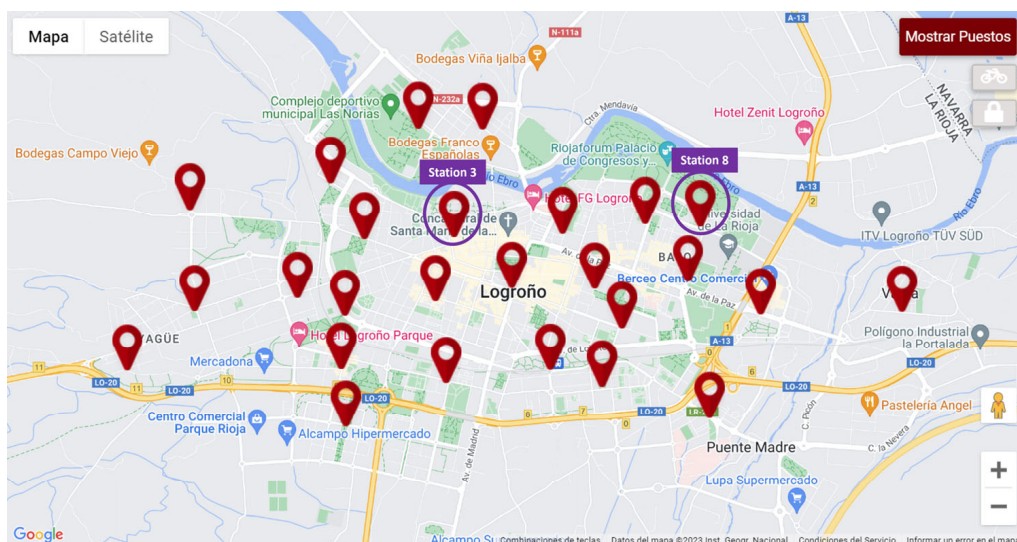

**Figure 2.** BiciLog stations distribution in the city of Logroño (Spain). Source: BiciLog website.

### 3.2. Data Processing

After cleaning the data, both datasets were merged using the User ID as a key. Moreover, additional variables needed for the analysis were added to the joined dataset, among them: month, year, starting and ending hour, and day of the week. The gender feature was created using both Is Man and Is Woman columns. The age of the participants was calculated using the date of birth. In order to study customers' behavior by group age, customers were divided into eight groups: (1) less than 10 years, (2) 10–19, (3) 20–29, (4) 30–39, (5) 40–49, (6) 50–59, (7) 60–69, and (8) more than 70.

To compute the distance travelled between the stations, the latitude and longitude of each station were used to compute the arc distance between each pair of stations. The arc distance corresponds to the shortest path between two stations. Therefore, it is only an estimation of the real distance travelled by users as with the information available it is not possible to know the real trajectory of each single trip. The trip duration was computed using the start and end time of each trip. Finally, features that were not needed were deleted from the dataset: User ID, Is maintenance user, Is Man and Is Woman. After cleaning, the final customer dataset had 3687 records while the trips file had more than 42k records. The final dataset variables are displayed in Table 2.

**Table 2.** Final dataset description.

| Trips and Customer Files Variables | Type of Variable |
| --- | --- |
| Start station ID | string |
| End station ID | string |
| Trip distance | numerical |
| Start date and time | datetime |
| End date and time | datetime |
| Trip duration | time |
| Hour | categorical |
| Day | categorical |
| Month | categorical |
| Year | categorical |
| Gender | categorical |
| Age group | categorical |

### 3.3. Methodology

The study performed a *t*-test for equality of means for the variables presented in Table 2. After the variables were collected and cleaned, the *t*-test was executed for sev-

eral combinations of pairs of variables. The statistical *t*-test allows for the analysis and comparison of different patterns of BSS usage dynamics. For instance, trips by different years, duration of rides, or customers' usage behavior, disaggregated by gender and age characteristics.

The *t*-test is a statistical hypothesis test used to check if there is a statistical difference between two groups. This test assumes the null hypothesis that the two groups considered are equal. If the null hypothesis is rejected, this indicates that both groups have a high probability of being different. There are different types of *t*-test depending on the groups under comparison and the direction of the test. This study performed a two-sample test, as in all cases the groups to be compared were considered as different populations. Moreover, the objective of this study is to identify if the groups are different, and therefore the two-tailed test was performed. The *t*-test values are expected to be significant between samples to conclude their mean statistical difference and prove the hypotheses of the study. To check the significance, the *t*-score (see Equation (1)) is compared against the Student's *t*-table to determine if the *t*-score is greater than the expected value, in order to reject the null hypothesis. The *p*-value can be also used to reject the null hypothesis if it is lower than the critical value considered. A critical value of 0.05 is generally accepted. More information can be found in [47].

$$t - score = \frac{\bar{Y}_1 - \bar{Y}_2}{\sqrt{\frac{s_1^2}{N_1} + \frac{s_2^2}{N_2}}} \tag{1}$$

where $N_1$ and $N_2$ are the sample sizes, $\bar{Y}_1$ and $\bar{Y}_2$ are the sample means, and $s_1^2$ and $s_2^2$ are the sample variances.

## 4. Empirical Results

BSS usage dynamics might be analyzed from different perspectives, among them the number of trips in a time interval. For example, in a day, the trip duration, and the trip's distance. For BiciLog, the analysis started by comparing the trips between different years. Table 3 displays different characteristics of trips made during 2021 and 2022 and shows interesting patterns. The results revealed that the daily number of trips in 2022 decreased in comparison with 2021. This is a surprising fact considering that after the release of COVID-19 mobility restrictions in Spain during 2021, the usage behavior was expected to increase as users could move freely around the city and also use bicycles to avoid crowded public transport. On the other hand, the trip duration in minutes considerably increased in 2022. BiciLog customers were doing on average 40 min longer trips than in 2021. Both trips by day and duration means are significantly different according to the *t*-test. The means of the distance between trips have not changed during the years considered, as is also corroborated by the *t*-test.

**Table 3.** *t*-test for equality of means for trips disaggregated by year.

|  | **2021** | **2022** | **Difference** | **t** | **p-Value** |
|---|---|---|---|---|---|
| Trips by day | 15.8 | 12.4 | −3.4 | 10.301 *** | $2.20 \times 10^{-16}$ |
| Trip duration (min) | 99.3 | 139.5 | 40.2 | −4.204 *** | $2.78 \times 10^{-5}$ |
| Trip distance (km) | 1.4 | 1.4 | 0.0 | −0.016 | 0.9866 |

Note: *** indicates statistical significance at the 1% level.

The gender gaps in BSS usage is a well-known topic in the BSS literature [48,49]. Moreover, this study is centered on understanding gender gaps to suggest specific campaigns to Logroño City Hall, to motivate bike usage among certain groups. Therefore, trips can be also studied with a gender perspective. As is displayed in Table 4, there are differences in how men and women are using the system, and all of them are significant according to the *t*-test. In general, women used the system less and made less trips during the day than men. These results are similar to what has already been found by previous studies, such as

the one conducted by Pellicer-Chenoll et al. [42]. However, women also made longer trips during the period considered. On average, women rent the bikes 7 min more than men. There are also differences between the distance travelled by both groups, and men travelled more than women. Nevertheless, it is important to consider that the distance value shown in this study is an estimation of the real distance travelled by BiciLog customers, as there is no possibility to know the real trajectory of every trip.

**Table 4.** *t*-test for equality of means for trips disaggregated by gender.

|  | Men | Women | Difference | t | *p*-Value |
|---|---|---|---|---|---|
| Trips by day | 15.6 | 12.6 | 3.0 | 8.803 *** | $2.20 \times 10^{-16}$ |
| Trip duration (min) | 115.9 | 123.0 | −7.1 | −0.741 | 0.458 |
| Trip distance (km) | 1.5 | 1.3 | 0.3 | 13.053 *** | $2.20 \times 10^{-16}$ |

Note: *** indicates statistical significance at the 1% level.

Users' distribution by gender and age is displayed in Figure 3. The youngest group (less than 10 years old) is also the smallest, with around 10 users. There are around 20 customers between 10 and 19 years old registered in BiciLog. The group of users more than 70 years old is also small, with less than 30 registered customers. The biggest group corresponds to customers between 20 and 29 years old, where the difference between men and women also becomes evident. Actually, the number of registered women in this age group is 1.5 more than men. In general, compared with men (around 1600), more women (around 2000) have signed up as BiciLog customers.

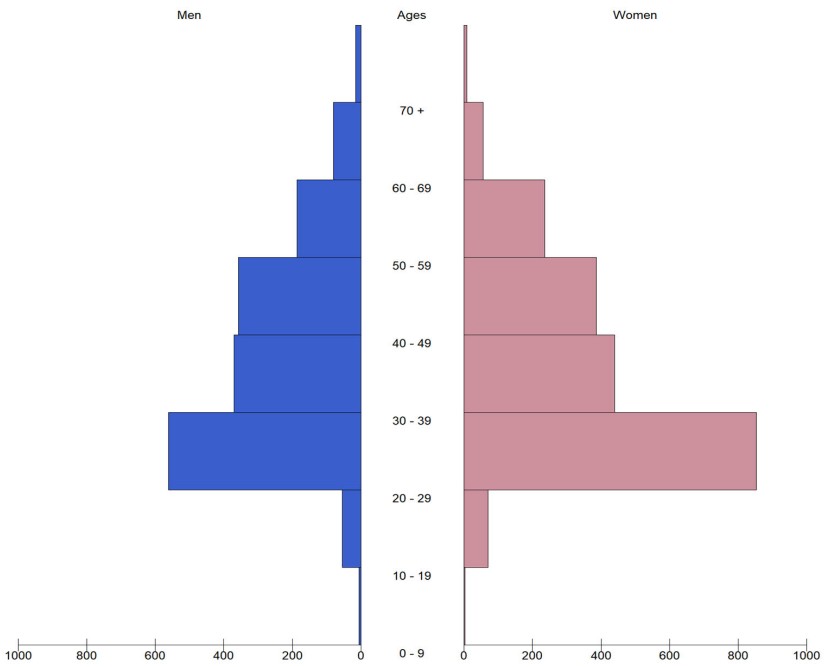

**Figure 3.** BiciLog customers by gender and age.

People under 20 years old and above 70 years old barely use the system. However, this is also related with their low proportion of registration, as is shown in Figure 3. The age groups with more registered users also make more daily trips. However, even when there are more women registered, they use the system less (2.3 daily trips on average) compared with men (2.9 daily trips in average), except women between 20 and 39 years old, as is shown in Table 5. A remarkable fact is that men between 60 and 69 years old have almost same levels of daily usage compared with men in their twenties. This result is similar to what was already found by Pans et al. [44]. In general, customers made 2.6 daily trips on average.

**Table 5.** *t*-test for equality of means for trips disaggregated by age.

|  | Men | Women | Difference | t | *p*-Value |
|---|---|---|---|---|---|
| 0–9 | 2.8 | 1.1 | 1.7 | 6.839 *** | $1.43 \times 10^{-7}$ |
| 10–19 | 2.1 | 1.5 | 0.5 | 2.249 ** | 0.02644 |
| 20–29 | 4.4 | 5.8 | −1.3 | −8.206 *** | $5.25 \times 10^{-16}$ |
| 30–39 | 2.7 | 3.0 | −0.3 | −3.032 *** | 0.002483 |
| 40–49 | 3.2 | 2.6 | 0.7 | 6.445 *** | $1.67 \times 10^{-10}$ |
| 50–59 | 3.2 | 2.2 | 1.0 | 9.646 *** | $2.20 \times 10^{-16}$ |
| 60–69 | 4.3 | 1.3 | 3.0 | 23.688 *** | $2.20 \times 10^{-16}$ |
| +70 | 1.2 | 1.1 | 0.1 | 1.296 | 0.2036 |

Note: *** and ** indicate statistical significance at the 1% and 5% levels, respectively.

A closer view of trip duration expressed in minutes (Table 6). shows that women make longer trips than men, except in the age groups from 20 to 29 and 60 to 69 years old. However, the time difference is significant only for those customers older than 40 years old. There are two hypotheses to explain the difference in trip duration: (i) Women use bikes as a means of entertainment, and (ii) women ride slower than men. Unfortunately, there is not enough information to test these hypotheses, but this provides new possibilities for the collaborators at Logroño's City Hall to improve BiciLog by taking into consideration customers' preferences by age and gender.

**Table 6.** *t*-test for equality of means for trip duration disaggregated by gender and age groups.

|  | Men | Women | Difference | t | *p*-Value |
|---|---|---|---|---|---|
| 20–29 | 134.9 | 113.6 | 21.4 | 1.200 | 0.230 |
| 30–39 | 111.6 | 149.8 | −38.2 | −1.560 | 0.119 |
| 40–49 | 80.0 | 131.3 | −51.2 | −2.832 *** | 0.005 |
| 50–59 | 83.6 | 136.5 | −52.9 | −2.970 *** | 0.003 |
| 60–69 | 214.3 | 84.6 | 129.7 | 6.244 *** | $1.25 \times 10^{-9}$ |

Note: *** indicates statistical significance at the 1% level.

Considering weekends and weekdays and gender, Table 7. shows there are no differences in usage during weekends when using the *t*-test (*p*-value equal to 0.71). Nevertheless, during weekdays, differences become more apparent, and as was mentioned before, men use more this BSS than women. Moreover, both groups—men and women—make more trips during weekdays than weekends. This could be related with the use of BSS for last-mile (or first-mile) connections to other transportation modes [50], especially during business hours.

Trip duration during weekdays displays significant differences for men and women (see *t*-test in Table 8); as mentioned before, women ride for longer times compared with men. Nevertheless, the ride's duration is similar for women and men during weekends. Additionally, there is not a significant difference in the trip's duration between weekdays and weekends for men nor for women.

**Table 7.** *t*-test for equality of means for weekend and weekday trips disaggregated by gender.

|  | Men | Women | Difference | t | *p*-Value |
|---|---|---|---|---|---|
| weekday | 17.6 | 13.6 | 4.1 | 10.667 | $2.20 \times 10^{-16}$ *** |
| weekend | 10.4 | 10.3 | 0.2 | 0.376 | 0.7071 |
| Difference | 7.2 | 3.3 |  |  |  |
| t | −15.513 | −7.539 |  |  |  |
| *p*-value | $2.20 \times 10^{-16}$ *** | $2.93 \times 10^{-13}$ *** |  |  |  |

Note: *** indicates statistical significance at the 1% level.

**Table 8.** *t*-test for equality of means for weekend and weekday trip duration disaggregated by gender.

|  | **Men** | **Women** | **Difference** | **t** | ***p*-Value** |
|---|---|---|---|---|---|
| Weekday | 113.3 | 122.8 | −9.5 | −0.757 *** | $4.49 \times 10^{-1}$ |
| Weekend | 122.2 | 123.5 | −1.3 | −0.104 | 0.917 |
| Difference | −8.8 | −0.7 |  |  |  |
| t | 0.942 | 0.047 |  |  |  |
| *p*-value | 0.347 | 0.963 |  |  |  |

Note: *** indicates statistical significance at the 1% level.

By analyzing Figure 4, we can confirm that the patterns found by Cortez et al. [34] about the most-used stations remain similar. According to these authors, the most-used stations in the BiciLog system are located in the city center. The most-used station, as mentioned in their study [34], is station number 3, and it is located in the city center (for reference, see Figure 2). Moreover, the main touristic places are around this station, as well as most of the music and drink spaces and restaurants (La Laurel Street). The Ebro Park is also nearby, and it is one of the biggest and most-visited parks in Logroño for leisure activities (see Figure 5a). Similarly, Station 8, called La Ribera, is located close to La Ribera Park and is surrounded by different sports facilities, schools and La Rioja University (see Figure 5b).

Checking the usage dynamics of the two most-used stations also provides interesting insights. For instance, at Station 3, there is no difference in the daily trips made by women or men between 30 and 49 years old. For the other age groups, there is difference in the number of trips made by women and men, according to the *t*-test results (see Table 9). The number of trips remains similar between customers in the considered age groups at Station 3. However, there are significant differences in the trip duration between customers of different ages, as is shown in Table 10. Customers between 40 and 49 years old make the shortest trips, while users under 40 years old ride for about an hour when they rent a bike from BiciLog. The most surprising fact is the duration of trips for customers between 60 to 69 years old. Trip duration for this age group is around 236 min (almost 4 h) for men and 127 min (more than 2 h) for women. This means that users in this age group make more than three-times-longer rides than other customers. None of the age groups show significant differences in trip duration between women and men, according to the *t*-test.

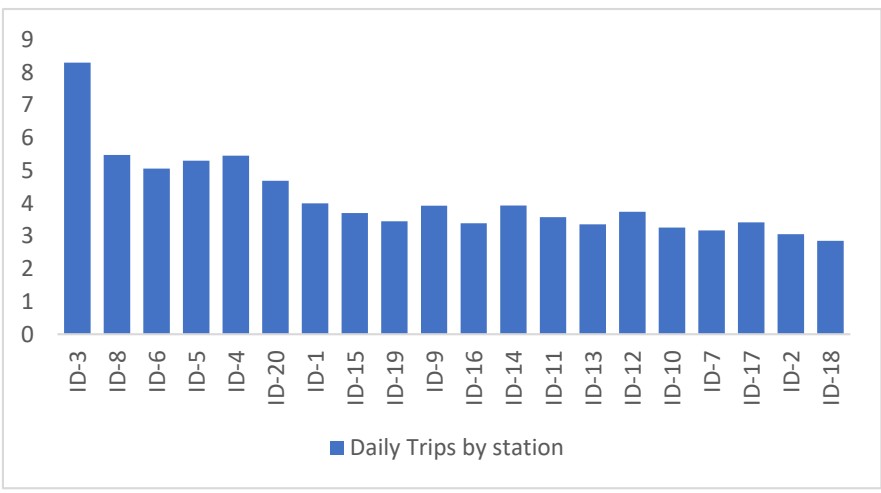

**Figure 4.** Trips disaggregated by station.

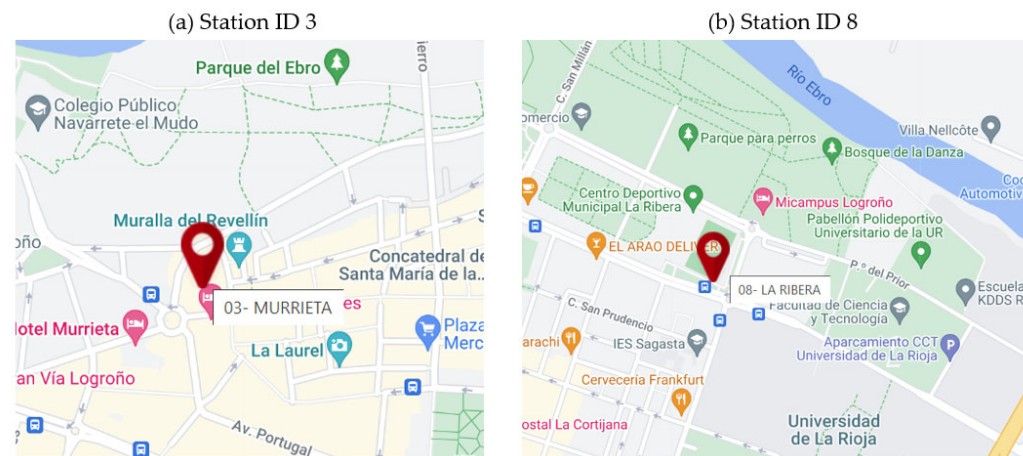

**Figure 5.** BiciLog station's location. (**a**) The most-used station, 3-Murrieta, is located in the city center and close to the main tourist areas, but it is also close to Parque del Ebro. (**b**) The second-most-used station, 8-La Ribera, is also near the city center and close to parks, schools, La Rioja University and sport facilities. Source: BiciLog website.

**Table 9.** *t*-test for equality of means for trips made at Station 3 disaggregated by gender and age group.

|  | **Men** | **Women** | **Difference** | **t** | ***p*-Value** |
|---|---|---|---|---|---|
| 20–29 | 2.7 | 3.3 | −0.6 | −3.898 *** | $1.38 \times 10^{-4}$ |
| 30–39 | 2.5 | 2.7 | −0.2 | −1.624 | 0.106 |
| 40–49 | 2.5 | 2.5 | 0.1 | 0.479 | 0.633 |
| 50–59 | 2.2 | 2.7 | −0.5 | −3.976 *** | $2.20 \times 10^{-4}$ |
| 60–69 | 3.2 | 2.0 | 1.2 | 21.482 *** | $2.20 \times 10^{-16}$ |

Note: *** indicates statistical significance at the 1% level.

**Table 10.** *t*-test for equality of means for the duration of trips made at Station 3 disaggregated by gender and age group.

|  | **Men** | **Women** | **Difference** | **t** | ***p*-Value** |
|---|---|---|---|---|---|
| 20–29 | 69.2 | 56.7 | 12.5 | 0.840 | 0.401 |
| 30–39 | 60.5 | 50.1 | 10.4 | 0.596 | 0.551 |
| 40–49 | 23.7 | 42.7 | −19.0 | −1.867 * | 0.063 |
| 50–59 | 45.5 | 76.5 | −31.0 | −1.647 | 0.102 |
| 60–69 | 236.6 | 127.8 | 108.8 | 1.594 | 0.116 |

Note: * indicates statistical significance at the 10% level.

Customers between 60 and 69 years old seem to be using the system as a mode of entertainment. As is displayed in Table 11, considering Station 3, there are significant differences in the duration and number of trips made by this age group and the other groups. These customers made more daily trips, and their rides are almost three times longer than the average rides in other groups. The privileged location of this station, close to shops, restaurants, bars and one of the main and biggest parks in the city, which as mentioned before has plenty of leisure activities, may have contributed to increasing the preference for this station, as well as to increasing the duration of rides among this age group.

**Table 11.** *t*-test for equality of means for trips made at Station 3 disaggregated by customers between 60 to 69 years old and other age groups.

|  | 60–69 | Others | Difference | t | *p*-Value |
|---|---|---|---|---|---|
| Trips | 3.1 | 2.7 | 0.3 | 4.78 *** | $1.97 \times 10^{-6}$ |
| Duration | 225.9 | 54.5 | 171.3 | 13.6 *** | $2.20 \times 10^{-16}$ |

Note: *** indicates statistical significance at the 1% level.

A similar analysis was performed at Station 8, the second-most-used station. As is shown in Table 12, there are no significant differences between the number of trips made by women or men for customers under 50 years old, but there are differences for the other considered groups, according to the *t*-test results. However, considering Station 8, the daily number of trips made by customers remains at similar levels to Station 3: between two and three daily trips by age group and gender. There are no differences in trip duration between women and men, as is shown in Table 13. Bike rides are also shorter than at Station 3 for all groups. At Station 8, customers ride for less than one hour and women between 50 and 59 years old rent bikes for longer time. However, as was already mentioned, there is no significant difference compared with the duration of trips made by men.

**Table 12.** *t*-test for equality of means for trips made at Station 8 disaggregated by gender and age group.

|  | Men | Women | Difference | t | *p*-Value |
|---|---|---|---|---|---|
| 20–29 | 2.9 | 2.9 | –0.1 | –0.291 | 0.772 |
| 30–39 | 3.3 | 2.5 | 0.8 | 2.728 *** | 0.009 |
| 40–49 | 2.7 | 2.3 | 0.4 | 1.726 * | 0.087 |
| 50–59 | 3.5 | 2.3 | 1.2 | 6.937 *** | $2.29 \times 10^{-10}$ |
| 60–69 | 2.6 | 2.0 | 0.6 | 9.441 *** | $2.20 \times 10^{-16}$ |

Note: *** and * indicate statistical significance at the 1% and 10% levels, respectively.

**Table 13.** *t*-test for equality of means for the duration of trips made at Station 8 disaggregated by gender and age group.

|  | Men | Women | Difference | t | *p*-Value |
|---|---|---|---|---|---|
| 20–29 | 38.5 | 42.1 | –3.6 | –0.278 | 0.781 |
| 30–39 | 58.6 | 32.5 | 26.0 | 1.395 | 0.168 |
| 40–49 | 47.5 | 39.6 | 8.0 | 0.502 | 0.616 |
| 50–59 | 45.6 | 136.0 | –90.4 | –1.697 * | 0.094 |
| 60–69 | 57.7 | 30.7 | 27.0 | 2.371 ** | 0.042 |

Note: ** and * indicate statistical significance at the 5% and 10% levels, respectively.

## 5. Discussion

In this study, an analysis of the usage patterns by age group and gender has been presented. The data used in this study are specific for the BiciLog system and the results cannot be reused directly for other cities, even if their BSS have similar characteristics (size, number of bikes and users, etc.). Nevertheless, some of the results found in this manuscript are in the same line as the results showed by other authors. For instance, women use less the system than men [42] or that older customers use more the system compared with younger customers [44]. A survey study could be useful to understand the motivation of customers to use the system and to corroborate the findings and conclusions made in this study. Other results, such as the impact of BSS usage on customers' health or the number of injuries, presented in a study by Woodcock et al. [4], cannot be analyzed with the current available information, but they can be considered for future work.

Some data have been dismissed during the cleaning phase. Data from 2020 were not considered as users' behavior was abnormal due to the COVID-19 mobility restrictions, and this would invalidate any analysis performed during that year. Five new docking stations

have been added to the system since the last quarter of 2021. This information represented less than 4% of the total dataset and was removed as there was not enough historical information of these stations to perform an historical analysis. In a similar way, garbled trips data (less than 0.5%) were removed from the dataset. Customers who lacked certain information, such as date of birth or gender, were not considered, as this information was key to performing this analysis. Deleting this information represents around 4% of the total dataset, and it could not be imputed as it is out of the scope of the right to privacy statement that is part of the GDPR laws. Removing the aforementioned data will not generate any bias. As many authors agree, less than 5% of missing data will not have any consequence on the quality of statistical inferences.

With the data available, the daily number of trips have been calculated, as well as the trip's duration and the travelled distance. Nevertheless, only an estimation of the travelled distance has been computed, as it is not possible to know the real trajectory of each single trip. Bicycles in BiciLog systems do not have a GPS incorporated and only the initial and final stations are reported. Therefore, the trip's distance between two stations was calculated using the Haversine formula, which computes the shortest distance between two stations considering the earth distance. This estimation could not always reflect usage route patterns, thus conclusions in this regard can be misled. In the future, if GPS is incorporated into the system, it could be interesting study usage routes to improve the service and its optimization.

As was shown in Table 3, the daily number of trips made in 2022 has decreased in comparison with 2021, even though the trip duration has increased by 40 min on average. At the same time, the estimated travelled distance has not changed during 2022. These factors could be explained as a transition of the BiciLog system from an alternative to public transportation modes or a replacement of last-mile (first-mile) [50] transport towards a means of entertainment, or to an element to improve the quality of life of inhabitants, as mentioned in Kwiatkowski's study [7]. Nevertheless, Kwiatkowski's results cannot be directly adapted to Logroño's case as the characteristics of this city and the communes analyzed in his study are different. Furthermore, a more in-depth user study will be needed to understand BiciLog usage patterns.

Previous studies, such as the one developed by Pans et al. [42], show that men are the main users in the Valencia BSS, but the results of this study showed that there are more women than men registered as BiciLog users. However, women use the system less than men, except for women between 20 and 39 years old. These findings are similar to those of previous studies [42–44]. In addition, women's trips are in general longer than men's, as was also concluded by Pans et al. [42]. Nevertheless, customers in the BiciLog system used the system extensively. Men's rides are around 115 min and women's are 123 min on average—a considerable difference compared with other BSS. For instance, in Valencia's BSS, men ride for 11.8 min and women for 12.1 min [42]. This could happen due to their use of BiciLog as an entertainment means. Unfortunately, with the available information, is not possible to test these hypotheses, but this could provide new possibilities to improve the BiciLog system according to customers' preferences.

The group of customers between 20 and 29 years old is the one that had more rides during 2021 and 2022. This is also the group with the most registered users. However, customers between 60 and 69 years old make a similar number of daily trips, but there are some remarkable differences. Similar to the results found by Pans et al. [42], older customers, in the BiciLog case those over 60, are using the system for longer rides, almost three times longer than other groups, as was shown in Table 11. This difference in trip duration could imply that this group of customers (+60 years old) uses the system as a mode of recreation. Moreover, the station where the ride's duration is more notorious—Station 3—is close to different shops, bars, restaurants, and El Ebro park, which is one of the biggest parks in Logroño and has plenty of leisure activities. Nevertheless, as mentioned before, with the current information, is not possible to test this hypothesis, and more information is needed to determine customers' preferences and motivations.

Considering the most-used stations (3 and 8), located in the city center, there are not significant differences between the number of trips taken by women and men by day, or the duration of the trips, in most of the age groups. The number of daily trips at both stations remains at similar levels, between 3 and 4, but customers at Station 3 also make longer trips than at Station 8. This could be related to the location of Station 3, located close to shops, bars, restaurants and one of the biggest parks in the city, where customers have plenty of leisure activities to enjoy. Another hypothesis is that stations located in the city center are better connected to other modes of transportation, while stations outside the city center do not ensure full cohesion with different transportations means, as was mentioned by Kwiatkowski [6]. Unfortunately, the efforts to integrate information from other public transport methods were unsuccessful and this hypothesis could not be tested. Finally, a limitation of this study is that the results are based on the information generated when customers use the system and not from direct feedback from BiciLog customers. To deeply understand the reasons behind the differences of how women and men use the system or how BiciLog is perceived and used according to age group, different surveys and information integration with other transportation modes are needed.

The policy implications of BSS are several, including the improvement in the user experience during bicycle trips, which also generates relevant social impacts on the population given the inclusion of age and gender in the bicycle program. In addition, BSS promote high flexibility, efficiency, and the implementation of new business models, as the sustainable operation of the system and its integration with public transport involves technological innovation in transport, which has contributed to a considerable change in the ways of owning and using any type of vehicle in recent years. Therefore, the final recipients are the authorities of the Logroño City Council, the citizens of Logroño, and the tourists of Logroño, since they have access to a competitive, ecofriendly, and efficient means of transport.

## 6. Conclusions

In this study, an analysis and exploration of BSS usage patterns by age and gender has been conducted. Data from a small system located in Logroño (Spain), called BiciLog, have been considered. The supervisors of this BSS are interested in exploring customers' data to understand the usage patterns of the system and make more informed decisions to motivate usage between certain groups. This way, they can improve this low-cost and environmentally friendly transportation mode and make the most of its health, environmental and economic benefits.

The *t*-test has been used to test the significance of the equality of means. The analysis of the data has shown that the number of trips reduced during 2022, which was an unexpected fact considering that COVID-19 mobility restrictions were lifted progressively during 2021 and the usage was expected to increase. However, the average of the trip's duration significantly increased during 2022. The first hypothesis considered in this study—that there are significant differences in BiciLog usage by gender—has been tested and confirmed, as was shown in the Results section. There are significant differences in the trips and their duration between women and men. In fact, even though women are the biggest group of registered customers, they took fewer daily trips than men. These results are in line with previous studies [42–44]. However, women usually ride for longer than men, which could imply they ride slower than men or that they use the BiciLog system as a mode of recreation instead of using it for the last-(or first-) mile, as is cited in the previous literature [50]. With the available information, it was not possible to determine the reason for women's longer rides. The second hypothesis—that there are significant differences between trips made by age groups and gender—is true to a certain degree. Citizens between 20 and 29 years old are the biggest group registered in the system and the ones who use it more. However, not all groups present differences, especially those with fewer users (less than 20 years old and over 70 years old). In general, women and men have different usage patterns according to their age, but some groups do not have differences. The analysis

conducted of the two most-used stations shows that there are no differences in how women and men in the same age groups are using the system (number of trips and duration), but there are differences between the age groups. These differences are notable for the users between 60 and 69 years old who extensively use the system, and for longer rides, as was also shown in previous studies [44]. The collaborators at Logroño City Hall can use this information to create and evaluate different campaigns to motivate and increase the use of this environmentally friendly and sustainable transport system among certain groups of users. However, a more in-depth study, including customers surveys, will be needed to understand their motivations and to corroborate the results found in this study.

The contribution of this study is an analysis of BSS customers' behavior patterns, and we disaggregated the examination by the gender and age group of customers. Logroño owns its BSS; however, a study of the characteristics of system´ users has not been developed in the city. Therefore, the City Hall authorities might not evaluate the effectiveness of the program. Supervisors might also want to focus on potential target groups to adapt the messaging of different campaigns and increase the number of customers who embrace this type of transport-sharing system, thus contributing, in the long term, to reducing air pollution and having a greener city.

There are several development lines that can be considered. The analysis can be extended to a longer period of data. Once enough data are available, it will be possible to test if the patterns found in this study can be replicated for other years. For example, the number of daily trips decreased in 2022, but the trips are longer than in 2021. Moreover, studying the patterns of new stations incorporated into the system is another possibility once sufficient information is collected. Another possibility is to perform a more in-depth study, including customer surveys, to understand and test the hypothesis raised in this study about the use of BiciLog as a mode of entertainment for certain customer groups. The incorporation of GPS devices into the bicycles can also help to better estimate users' preferred routes. Real-time reports would be possible with permanent access to the original sources and by setting up a server to handle data.

Finally, there are some considerations and challenges for BSS. The task of re-distributing bicycles across cities might be solved by installing GPS on shared bicycles, which provides real-time tracking, a rebalancing of sources, and data to design transport plans. BSS can be expanded to new segments of the population, which also reduces the barriers linked to trip length and excessive heat. BSS apply different methodologies depending on the country or city. Therefore, it is necessary to design a standard methodology to measure the impacts of BSS in terms of climate change, congestion, air pollution, noise quality, health of users, and time savings.

**Author Contributions:** The authors contributed extensively to the work presented in this paper. Writing-original draft preparation, A.C.-O.; writing-review and editing, A.B.T.-P. All authors have read and agreed to the published version of the manuscript.

**Funding:** We extend our gratitude and acknowledgment to the Universidad de las Américas, which financially supported this research (2023).

**Institutional Review Board Statement:** Not applicable.

**Informed Consent Statement:** Not applicable.

**Data Availability Statement:** The datasets used and analyzed in this study are property of Logroño City Hall and authors do not have permission to share the original files.

**Acknowledgments:** We reveal special remarks to Logroño City Hall members, which provide data for the analysis presented in this study.

**Conflicts of Interest:** The authors declare no conflict of interest. The funders had no role in the design of the study; in the collection, analyses, or interpretation of data; in the writing of the manuscript, or in the decision to publish the results.

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
