# Peer review of "The Effect of Gender and Age in Small Bicycle Sharing Systems: Case Study from Logroño, Spain"

_sustainability, doi:10.3390/su15107925_

Round 1

Reviewer 1 Report

The article touched upon a topic that is somewhat relevant in our time. In general , not bad for the first option.

But it seems that for publication in many MDPI journals, including the Sustainability journal, some improvements need to be made.

Comments:

1.     Tell us about the model used in the abstract.

2.     Give this part in the abstract in the introduction.

During the last years, bike sharing systems (BSS) have been adopted in many large cities around the world. Thanks to their environmental and health benefits, BSS popularity as a green transportation mode is exponentially increasing and many small cities are also adopting them. However, few of these small cities count with the resources to manage and analyze the massive amount of data produced by these systems in order to optimize them and promote their use among citizens.

3.     Remove these parts from the introduction and give them in Conclusion and political recommendations.

The results of the study show that the quantity of daily trips have re­duced during 2022 compared with 2021 but the trips made are longer. Moreover, there are more women than men registered on BiciLog, but women use less the system than men even when they ride for longer time. Moreover, citizens between 20 and 29 years old are the biggest group registered in the system and the ones who use it more. Nevertheless, customers between 60 and 69 years old extensively borrow bikes and ride longer time which could imply that they use the system as an entertainment mean instead of as a mode of connection to other transportations forms. These results can give supervisors initial in­sights to establish campaigns to promote and increase the use of this environmentally friendly and sustainable transport system among certain population groups. However, a more in-depth study, including customers surveys, will be needed to understand their motivations and to corroborate the results find in this study, The contribution of this study is an analysis of BSS customer's behavior patterns, dis­aggregating the examination by gender and group age of customers. Logrono owns its BSS; however, a study of the characteristics of system' users has not been developed in the city. Therefore, the City Hall authorities might not evaluate the effectiveness of the program. The results found in this investigation about the usage patterns among different gender or age groups can help supervisors of BiciLog to make appropriate decisions to promote the usage of BiciLog in the city. Besides, supervisors can use this information to create and evaluate different campaigns dedicated to improving the system making it safer and more comfortable for their customers. Finally, supervisors might also want to focus on potential target groups to adapt the messaging of different campaigns and in­crease the number of customers who embrace this type of transport sharing systems con­tributing, in the long term, to reduce the air pollution and have a greener city.

4.     Enter this section from Theory and hypotheses there by creating a literature review section.

Prior studies analyzed BSS using different disaggregation according to the research objectives. (Beecham et al., 2014) study the commuting behaviors of users by analyzing the journey data to gain insights on the people using those services, and the geography of commuters' workplaces. Specifically, BSS usage by gender and group age generated dif­ferent findings. Women use the BSS less than men in all the tested time slots of the study performed in Valencia (Pellicer-Chenoll et al., 2021), which also showed different network density and centrality between genders, concluding the presence of gender inequality pat­terns of use and space occupation. Moreover, a study performed in the largest public BSS in the U.S. (Bluebikes in Boston, City Bike in New York, and Divvy Bikes in Chicago) showed that the portion of trips made by women increased over 22% in all programs in the period of 2014-2018 (Hosford & Winters, 2019). However, in all schemes, the gender gap was higher for older bicycle share users. (Pans et al., 2023) indicated that women use Valencia BSS less than men and older people employ the BSS more frequently than younger people, while people with lower socio-economic level have a higher rate usage than people with a high and medium socio-economic level. Moreover, a study performed in Paris Region-Petite Couronne (France) established that women used BSS much less than men (about 30% of the total number of users) (Gorrini et al., 2021). Women are concerned about: accessibility (availability of bikes, distance to the nearest station, type and quality of the cycle paths), safety and security, social constraints, weather, and topography. Fi­nally, women were most affected by COVID 19 pandemic as they are more likely to be dependent on public transportation, which was heavily restricted during 2020 and 2021. Particularly, a BSS study conducted in Lisbon (Teixeira & Cunha, 2023) revealed that men have higher bike ownership rates and combine the personal use of bike with BSS. Con­versely, women more frequently integrated BSS and public transport. Furthermore, while men were using BSS more regularly than women pre-pandemic, during COVID 2019 pan­demic women were using BSS as recurrently as men.

5.     Improve methodological explanation.

6.     Please also use sufficiently published articles in many of MDPI's journals, including Sustainability journal, that are relevant to the topic of your article. The appeal to these articles is justified both from an ethical and logical point of view. 

I think the remarks will not cause you any difficulties

Good luck to you

Author Response

In the attached document, you can find the response to reviewers. Thank you!

Reviewer 2 Report

The authors of the manuscript presented a very interesting and current topic. I believe that the presented research should be continued and extended. The topic is original and relevant in the field.

The methodology was well presented. The research process is logical.

There are interesting figures in the manuscript - well enriching the text. This is the strength of the manuscript.

The manuscript is worth publishing after the corrections have been made.

Detailed remarks:

-       The number of words in the abstract exceeds the limit of 200.

-       There is a reference error in the text: Error! Reference source not found.

Author Response

(The authors gave the same response as above.)

Reviewer 3 Report

The article is an interesting solution to the research problem, but it is based solely on a case study. The analysis of a specific example does not necessarily lead to general conclusions that can be implied to other examples. This is the first and main reservation that the article does not compare, for example, two research groups in different places. This is a typical description of a specific example. Such studies are valuable but have limited scientific impact.

The article concerns the analysis of behavior and communication preferences regarding the use of the bike sharing system due to age and gender. It is a valuable study in this respect because there are no scientific studies focused in this way.

Below I present in points my comments, reservations and suggestions for improving the article.

1. At the beginning of the article, a synthesized paragraph of description about the research site is missing. There should be a map here (in accordance with the rules of cartography, i.e. with a legend and scale) showing what place the article is about and where it is located. Logrono is a small town and not every reader will know where it is.

2. The article is based on a relatively small number of scientific sources, and it should. I strongly encourage the authors to analyze the texts more extensively, then the article will have much more value. A structured review of the literature on the functioning of bikesharing systems and research on user preferences and behavior is essential.

3. Examples of scientific studies:
https://www.mdpi.com/1996-1073/13/23/6240/htm/
https://www.sciencedirect.com/science/article/abs/pii/S136192092100184X
https://www.sciencedirect.com/science/article/abs/pii/S2213624X21000390
https://apcz.umk.pl/BGSS/article/view/35339

4. The structure of the article is generally correct. It is nice to separate the discussion and the summary, but I think that in these places the own results are too weakly embedded in relation to other studies. The discussion should definitely be sharpened by referring your own results to other studies, especially since this is a study based on a single case study.

5. The way content is represented in the Figure 3 illustrations says nothing. This way is illegible and irrelevant. You should present a map of the city, locate points, or zoom in on the area of interest.

Author Response

(The authors gave the same response as above.)
